# Reassessment of the involvement of Snord115 in the serotonin 2c receptor pathway in a genetically relevant mouse model

Jade Hebras[1†], Virginie Marty[1†], Jean Personnaz[2], Pascale Mercier[3], Nicolai Krogh[4], Henrik Nielsen[4], Marion Aguirrebengoa[5], Hervé Seitz[6], Jean-Phillipe Pradere[2], Bruno P Guiard[7], Jérôme Cavaille[1]*

[1]Laboratoire de Biologie Moléculaire Eucaryote, Centre de Biologie Intégrative, Université de Toulouse, CNRS, UPS, Toulouse, France; [2]Institut National de la Santé et de la Recherche Médicale (INSERM), U1048, Institut National de la Santé et de la Recherche Médicale (INSERM), France Institut des Maladies Métaboliques et Cardiovasculaires (I2MC), Université de Toulouse Université Paul Sabatier, Toulouse, France; [3]Institut de Pharmacologie et de Biologie Structurale (IPBS), Centre National de la Recherche Scientifique UMR5089, Toulouse, France; [4]Department of Cellular and Molecular Medicine, University of Copenhagen, Copenhagen, Denmark; [5]LBCMCP, Centre de Biologie Intégrative (CBI), CNRS, Université de Toulouse, Toulouse, France; [6]IGH (CNRS and University of Montpellier), Montpellier, France; [7]Centre de Recherches sur la Cognition Animale (CRCA), Centre de Biologie Intégrative (CBI), Centre National de la Recherche Scientifique, Université de Toulouse, Toulouse, France

*For correspondence:
jerome.cavaille@univ-tlse3.fr

[†]These authors contributed equally to this work

Competing interests: The authors declare that no competing interests exist.

**Abstract** *SNORD115* has been proposed to promote the activity of serotonin (HTR2C) receptor via its ability to base pair with its pre-mRNA and regulate alternative RNA splicing and/or A-to-I RNA editing. Because *SNORD115* genes are deleted in most patients with the Prader-Willi syndrome (PWS), diminished HTR2C receptor activity could contribute to the impaired emotional response and/or compulsive overeating characteristic of this disease. In order to test this appealing but never demonstrated hypothesis in vivo, we created a CRISPR/Cas9-mediated *Snord115* knockout mouse. Surprisingly, we uncovered only modest region-specific alterations in *Htr2c* RNA editing profiles, while *Htr2c* alternative RNA splicing was unchanged. These subtle changes, whose functional relevance remains uncertain, were not accompanied by any discernible defects in anxio-depressive-like phenotypes. Energy balance and eating behavior were also normal, even after exposure to high-fat diet. Our study raises questions concerning the physiological role of *SNORD115*, notably its involvement in behavioural disturbance associated with PWS.

## Introduction

Box C/D small nucleolar RNAs (SNORDs) represent a well-defined family of small non-coding RNAs that exert their regulatory functions via antisense-based mechanisms. Indeed, the vast majority forms base-pairing interactions with ribosomal RNA (rRNA) precursor, spliceosomal U6 snRNA or tRNA and, in doing so, they guide sequence-specific ribose methylations (*Cavaillé et al., 1996*; *Ganot et al., 1999*; *Kiss-László et al., 1996*; *Vitali and Kiss, 2019*) or, in rare cases, base acetylation (*Sharma et al., 2017*). By acting as RNA folding chaperones, a few are specialized to facilitate

the production of mature 18S, 28S, and 5.8S rRNA (*Watkins and Bohnsack, 2012*). Finally, nucleoli also contain orphan SNORDs lacking conserved antisense elements against canonical RNA targets, thus raising the possibility that some may interact with uncharacterized targets, possibly including mRNAs (*Bratkovič et al., 2020*).

Intriguingly, many orphan SNORDs are generated from two large, imprinted chromosomal domains at human 15q11q13 and 14q32 (*Cavaillé, 2017*; *Marty and Cavaillé, 2019*). The imprinted human 15q11q13 region - also known as the Prader-Willi Syndrome (PWS)/Angelman Syndrome (AS) locus or SNURF-SNRPN domain - contains several paternally expressed, protein coding genes as well as numerous paternally expressed, neuronal-specific SNORD genes organized as two main repetitive DNA arrays: the *SNORD116* and *SNORD115* clusters composed of 29 and 47 related gene copies, respectively (*Cavaillé et al., 2000*). The lack of expression of one (or several) paternally expressed gene(s) on 15q11q13 is genetically linked to Prader Willi Syndrome (*Buiting, 2010*; *Cassidy et al., 2012*). This rare human genetic disorder is characterized by poor feeding and muscular hypotonia in the perinatal period while, in the childhood and adulthood, individuals with PWS manifest an obsession with food and eating that, if not controlled, leads to life-threatening obesity. Other symptoms include decreased energy expenditure, growth delay, abnormalities in hormonal profiles, hypogonadism, dysfunction of the hypothalamus-pituitary axis, respiratory problems, high pain threshold, sleep disturbance, learning disabilities as well as diverse psychopathological issues notably in terms of obsessive-compulsive behavior and anxio-depressive traits (*Butler et al., 2019*; *Cassidy et al., 2012*; *Muscogiuri et al., 2019*; *Thuilleaux et al., 2018*).

Although the genotype-phenotype relationship in PWS is complex and not entirely understood, deficiencies in SNORDs have recently emerged as one of the key determinants for the disease. This is evidenced by rare PWS individuals bearing ~71–200 kb-long paternally inherited deletions overlapping the SNORD gene array, particularly *SNORD116* genes (*Bieth et al., 2015*; *de Smith et al., 2009*; *Duker et al., 2010*; *Fontana et al., 2017*; *Sahoo et al., 2008*; *Tan et al., 2020*). Remarkably, *Snord116*-deficient mice are growth-retarded and show increased food intake, two abnormalities reminiscent of PWS phenotype (*Ding et al., 2008*; *Skryabin et al., 2007*). The relevance of increased food intake with respect to the lower body weight of *Snord116* mutants was, however, recently questioned (*Polex-Wolf et al., 2018*). Importantly, loss of *Snord116* is not accompanied by any signs of obesity and, paradoxically, *Snord116*-deficient mice appear even protected from high fat-diet induced obesity (*Ding et al., 2008*; *Qi et al., 2016*). Thus, *Snord116*-KO mice, like all other PWS mouse models, do not fully recapitulate the entire set of PWS phenotypes (*Bervini and Herzog, 2013*).

The *SNORD115* gene locus has attracted particular attention due to its predicted involvement in promoting the synthesis of serotonin HTR2C receptor with optimal signaling properties (*Cavaillé, 2017*; *Cavaillé et al., 2000*; *Stamm et al., 2017*), very likely by fine-tuning alternative RNA splicing and/or adenosine (A)-to-inosine (I) RNA editing of *HTR2C* pre-mRNA (*Kishore and Stamm, 2006*; *Vitali et al., 2005*). Indeed, a growing body of observations supports a functional link between *SNORD115*, HTR2C, and PWS. First, adult mice bearing a constitutive inactivation of the *Htr2c* gene are hyperphagic and overweight (*Nonogaki et al., 1998*; *Tecott et al., 1995*). Second, increased food intake and growth retardation were reported in knock-in mice that only produce the fully edited (VGV), less active HTR2C isoform (*Kawahara et al., 2008*; *Morabito et al., 2010a*). Third, induced changes in *Htr2c* alternative splicing patterns mediated by antisense oligonucleotides alter food intake in mice (*Zhang et al., 2016*). Fourth, defects in HTR2C pathways, including altered RNA editing, are involved in the pathophysiology of psychiatric disorders, particularly anxiety, depression, obsessive-compulsive behaviors, and schizophrenia (*Chagraoui et al., 2016*). Fifth, and perhaps even more tellingly, defects in *Htr2c*-related behaviors were described in the PWS-IC mouse model in which all paternally inherited alleles in the cluster are silenced (*Davies et al., 2019*; *Doe et al., 2009*; *Garfield et al., 2016*). Altogether, defective post-transcriptional processing of *HTR2C* pre-mRNA - and thus impaired HTR2C pathway - has been surmised to account for overeating and anxio-depressive phenotypes frequently encountered in PWS patients (*Cavaillé, 2017*; *Stamm et al., 2017*), although the loss of *SNORD115* is unlikely to be sufficient to elicit PWS (*Bürger et al., 2002*; *Runte et al., 2005*). This model was first proposed 20 years ago, and it constitutes certainly the most convincing example advocating for an involvement of SNORD in modifying a protein-coding transcript (*Cavaillé et al., 2000*). Yet, direct proofs supporting this appealing *SNORD115*/HTR2C axis are still lacking at the organism level (*Cavaillé, 2017*). As a corollary,

whether lack of SNORD115 per se is sufficient to trigger altered HTR2C-signaling pathways and, in doing so, leads to behavioral abnormalities remains to be formally demonstrated. Hence, probing the mode of action of *SNORD115* in vivo is not only a prerequisite to further our understanding of the pathophysiology of PWS but it is also of immediate interest to uncover novel SNORD-mediated functions in the brain.

Here, we generated a CRISPR-Cas9-mediated specific *Snord115* gene knockout (KO) mouse model to better understand the functional relevance of *Snord115* in influencing *Htr2c* pre-mRNA processing and HTR2C-related behaviors (e.g. eating and emotional response). We unexpectedly showed that constitutive deletion of the entire *Snord115* gene array does not greatly alter the post-transcriptional regulation of *Htr2c* pre-mRNA, nor does it affect emotionality, homeostatic feeding or energy balance, either on normal chow or high-fat diets. In contrast to common belief, we thus conclude that a constitutive loss of *Snord115* has a relatively modest impact on two well-known HTR2C-mediated physiological processes.

## Results

### CRISPR-Cas9-mediated disruption of the *Snord115* gene array

*Snord115* is processed from repeated, regularly spaced introns of a poorly characterized noncoding RNA host-gene transcript (*Figure 1A*). *Snord115* sequences are the only conserved DNA segments in placental mammals, strongly arguing that this atypical transcription unit encodes functional small RNA species under selective pressure (*Cavaillé et al., 2000*). In order to probe the physiological relevance of *Snord115* in vivo, we engineered a KO mouse model carrying a targeted, constitutive CRISPR/Cas9–mediated deletion removing the entire *Snord115* gene array (*Figure 1A–B*). Because *Snord115* genes are only expressed from the paternal allele, ~50% of *Snord115*-deficient mice are expected to be obtained in the progeny after crossing wild-type females with heterozygous males. Indeed, the *Snord115* gene array is no longer expressed upon a paternally inherited deletion while, as expected, its expression remains unaffected upon maternal transmission of the deletion (*Figure 1C*). Moreover, we were also unable to detect *Snord115* expression by RT-qPCR in several manually-dissected brain structures from adult mice bearing a paternally-inherited deletion (*Figure 1D*). Importantly, the removal of *Snord115* genes does not significantly alter the expression of the neighboring imprinted genes (*Figure 1E*), including that of the anti-*Ube3a* transcript thought to silence the paternal *Ube3a* allele in cis (*Figure 1F*). Altogether, these important controls render unlikely the possibility that we removed critical regulatory regions controlling imprinted expression over the domain. Finally, in order to limit confounding effects due to genetic heterogeneity and/or undesired CRISPR-Cas9-mediated cleavages, if any, we also backcrossed to C57BL/6J background for at least eight generations before proceeding to phenotypic exploration. The term *Snord115*-deficient or *Snord115*-KO mice will thus be used to refer to heterozygous mice with a paternally inherited deletion. *Snord115*-deficient mice are seemingly normal and progeny was obtained at expected Mendelian and sex ratios, leaving unlikely any developmental defect and/or perinatal lethality.

### The repertoire of 2'-O-methylations of ribosomal RNAs remains unchanged in the adult brain of *Snord115*-deficient mice

*Snord115* is a massively-expressed, neuronal-specific SNORD that stably associates with the 2'-O-methyltransferase Fibrillarin in vivo and accumulates in the nucleolus (*Bortolin-Cavaillé and Cavaillé, 2012*; *Cavaillé et al., 2000*). Yet, it lacks an antisense element that can guide Fibrillarin to rRNA targets, arguing against a direct role in specifying neuronal-specific rRNA 2'-O-methylation(s). In agreement with this assumption, our recent study failed to identify any brain-specific methylation rRNA sites (*Hebras et al., 2020*). We nevertheless hypothesized that high *Snord115* levels in neurons may impact on rRNA methylation profiles, possibly by modulating the availability of other SNORD-associated binding proteins. To address this, RiboMeth-seq was applied to adult whole brains from WT and *Snord115*-deficient mice. As reported in *Figure 1G*, the overall rRNA methylation patterns were comparable between WT and *Snord115*-KO brain, with only slight changes in methylation levels at a few rRNA sites: SSU-A576, LSU-C1868, and 5.8S-U14. Since methylation levels at these three positions were unaffected in the developing (E16.5) brain of another mouse model where *Snord115* and

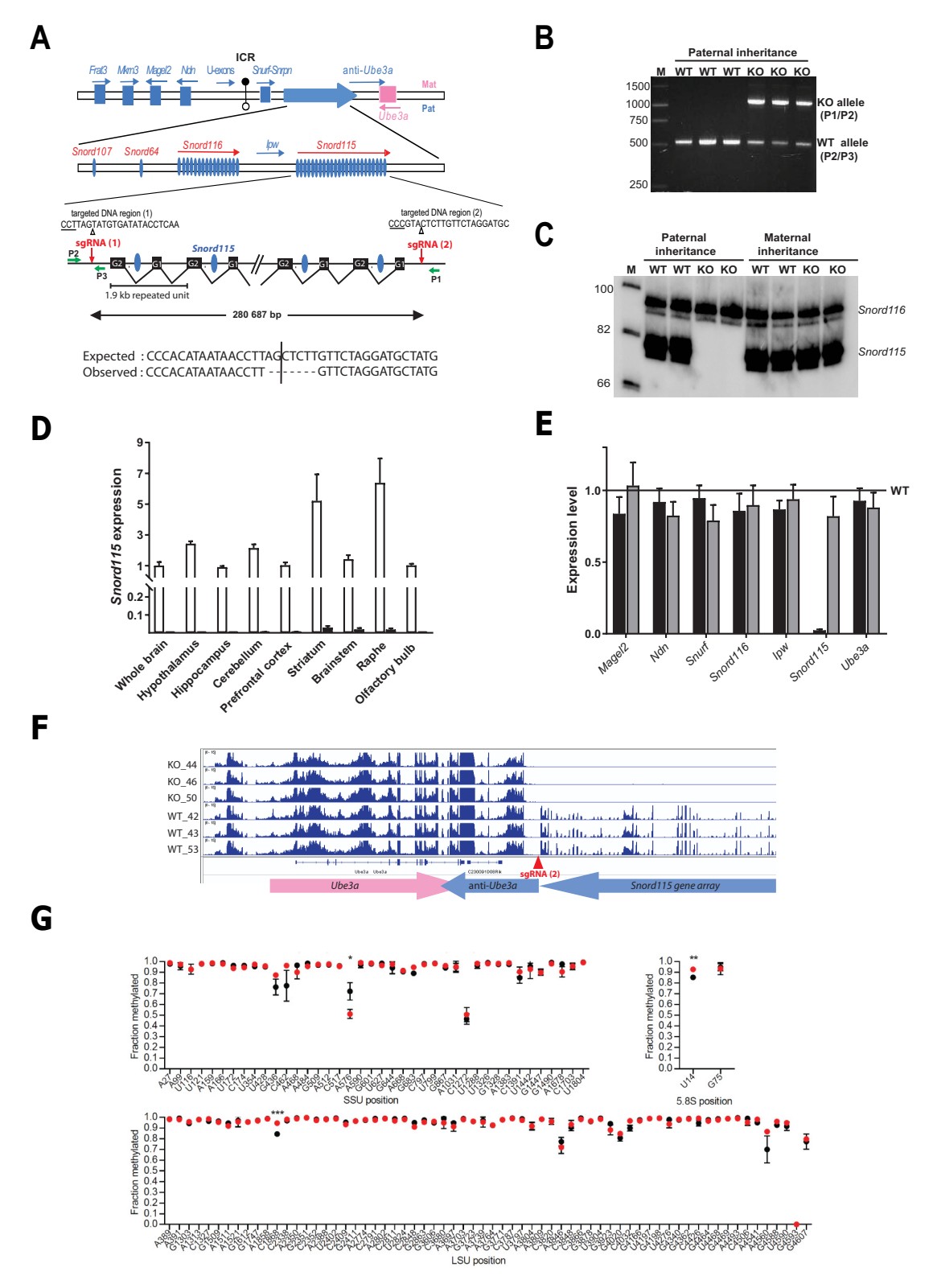

**Figure 1.** Targeted deletion of the paternally expressed *Snord115* genes. (**A**) Schematic representation of the imprinted *Snurf-Snrpn* region on mouse chromosome 7 c, also commonly known as the imprinted PWS region at human 15q11q13. Paternally expressed and maternally expressed protein-coding genes are symbolized by blue and pink rectangles, respectively. The paternally expressed SNORD gene array composed of *Snord107*, *Snord64* and the tandemly repeated *Snord116* and *Snord115* gene families is depicted by a large, horizontal blue arrow while poorly defined noncoding

*Figure 1 continued on next page*

*Figure 1 continued*

transcript (U-exons, anti-*Ube3A* and *Ipw*) are denoted with thin arrows. SNORD gene copies are denoted as vertical blue ovals. The Imprinting Centre Region (ICR) that controls imprinted expression over the domain is represented by filled and open lollipops (methylated and un-methylated alleles, respectively). Each *Snord115* gene copy is intron-encoded within a ~ 1.9 kb tandemly- repeated unit spreading over a ~ 280 kb-long noncoding DNA region. The two targeted DNA sequences by sgRNA(1) and sgRNA(2), as well as the theoretical and observed editing events at the deletion junction, are shown. The PAM sequence is underlined and the position of the predicted cleavage sites are indicated (positions 17 and 18 of the sgRNA). The genomic coordinates of the expected deletion is mm10 chr7:59,340,581–59,621,267. Green arrows represent the relative positions of P1, P2, and P3 primers used for PCR genotyping. Scale is not respected. (B) PCR genotyping of WT and mutant mice carrying a paternally-inherited deletion using a mixture of P1, P2, and P3 primers. (C) Expression of *Snord115* and *Snord116* was assayed (northern blot) in whole brains from heterozygous mice having a paternally- or maternally-transmitted deletion and WT littermates (n = 2 individuals per genotype). (D) Black and white bars represent *Snord115* expression (RT-qPCR relative to *Gapdh*) in several brain areas dissected from mice bearing a paternally inherited deletion and WT littermates, respectively (n = 8–12 per genotype). (E) Black and gray bars represent expression levels (RT-qPCR relative to *Gapdh*) of flanking imprinted genes in the whole brain of mice bearing a paternally and maternally inherited deletion, respectively (n = 4–6 per genotype). Expression levels of WT were arbitrarily set to 1. (F) IGV screenshot showing that a targeted deletion of *Snord115* gene array does not perturb the expression at the *Ube3A*/anti-*Ube3A* gene locus (mRNA-seq datasets generated from hypothalamus; *supplementary file 3*). (G) Methylation levels (RiboMeth-seq) at 2'-O-methylated rRNA sites in WT (black circles) and *Snord115*-deficient (red circles) adult brain (n = 3 per genotype). Nucleotide numbering is according to human rRNA (*Krogh et al., 2016*). Asterisks indicate statistical significance levels.

The online version of this article includes the following figure supplement(s) for figure 1:

**Figure supplement 1.** Loss of *Snord116* and *Snord115* does not impact on rRNA methylation profiles in the adult brain.

*Snord116* genes are simultaneously deleted (*Figure 1—figure supplement 1* and *supplementary file 1*), we thus conclude that neither *Snord115*, nor *Snord116* play a prominent role in ribose methylation of rRNAs.

## *Snord115* does not influence alternative RNA splicing of *Htr2c* pre-mRNA

The *Htr2c* pre-mRNA is alternatively spliced through the use of two 5' donor splice sites (5'-SS) (*Figure 2A*). The regular 5'-SS generates RNA2 encoding the full-length receptor while a cryptic 5'-SS yields an alternatively spliced transcript (RNA1) that generates a truncated, presumably non-functional receptor. The binding of *Snord115* nearby the cryptic 5'-SS was proposed to favor the production of RNA2 (*Cavaillé et al., 2000*; *Doe et al., 2009*; *Garfield et al., 2016*; *Kishore and Stamm, 2006*). According to this model, the RNA2-to-RNA1 ratio should decrease in the absence of *Snord115*. As shown in *Figure 2B*, the steady state levels of *Htr2c* mRNAs, as assayed by a couple of primers matching the 5'-UTR, were globally unchanged in whole brain as well as in eight manually-dissected brain regions of adult *Snord115*-deficient mice. Only striatum showed a significant increase in *Htr2c* mRNA abundance, as compared to WT. We then tested the lack of *Snord115* on alternative splicing of *Htr2c* pre-mRNA by measuring the RNA2-to-RNA1 ratio by RT-qPCR. Again, lack of *Snord115* did not cause any obvious changes in the profiles of alternative splicing, including in spinal cord (*Figure 2C* -left). Adult hypothalamus even displayed a slight increase in the RNA2-to-RNA1 ratio. This effect was, however, not seen in another independent cohort (not shown). As controls, and in agreement with previous findings (*Canton et al., 1996*), choroid plexus showed highest RNA1 level as compared to hypothalamus (*Figure 2C* -right), thus indicating that our RT-qPCR procedure is capable of detecting changes in the RNA2-to-RNA1 ratio. Finally, we demonstrated unaltered alternative splicing of hypothalamic *Htr2c* pre-mRNAs by RT-PCR analysis with a pair of primers that simultaneously detected RNA2 and RNA1 (*Figure 2D*). Thus, our findings do not support a central role of *Snord115* in modulating alternative splicing of *Htr2c* pre-mRNA in vivo.

## The lack of *Snord115* has a weak impact on RNA editing of *Htr2c* pre-mRNA

Through the action of ADAR2 and ADAR1, *Htr2c* pre-mRNA undergoes site-specific A-to-I RNA editing at five adenosines clustered in exon V: the A-, B-, C-, D-, and E- sites (*Figure 3A* -top; *Burns et al., 1997*). Strikingly, *Snord115* is predicted to target the C-site for 2'-O-methylation (*Figure 2A*; *Cavaillé et al., 2000*). Inosines are interpreted as guanosines during translation and, accordingly, these nucleotide changes result in alteration of three amino acids positioned within the second intracellular loop of the receptor. The protein recoding events lead to the production of 24

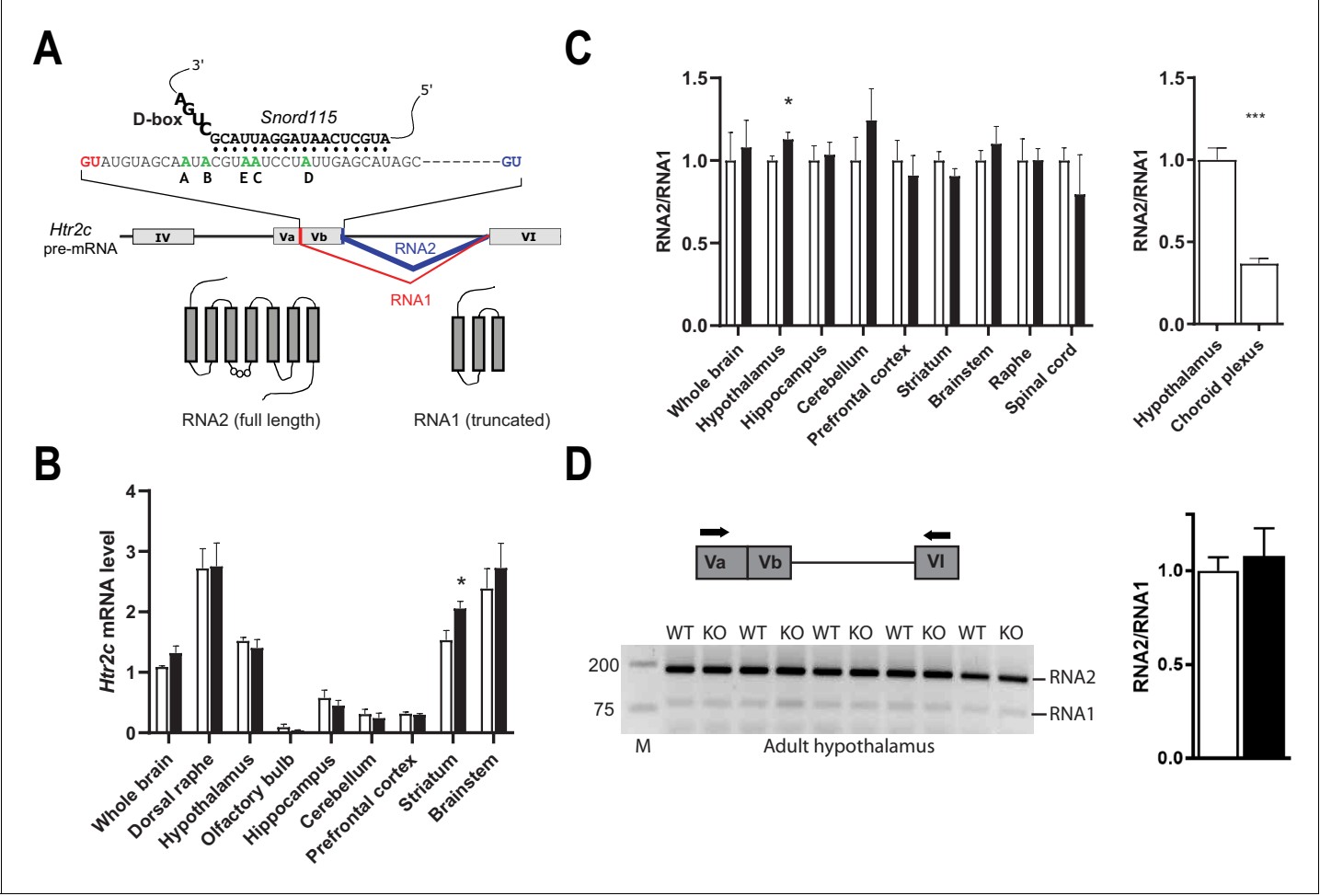

**Figure 2.** Alternative RNA splicing of *Htr2c* pre-mRNA remains unaffected in *Snord115*-deficient brains. (**A**) Through the use of regular (blue) and alternative (red) 5' splice sites, RNA splicing of *Htr2c* pre-mRNA generates RNA2 and RNA1 transcripts that encode full length and truncated HTR2C receptor, respectively. *Htr2c* pre-mRNA undergoes A-to-I RNA editing at five adenosines (green), denoted as the A, B, C, D, and E editing sites. *Snord115* has the potential to target the C-site for 2'-O-methylation, that is the position paired to the fifth nucleotide upstream of the D-box (CUGA). (**B**) Black and white bars represent *Htr2c* mRNA expression levels (RT-qPCR relative to *Gapdh*) in whole brain and manually dissected brain regions of *Snord115*-deficient mice and WT littermates, respectively (n = 5–8 per genotype). (**C**) Left: Black and white bars show the RNA2-to-RNA1 ratio (RT-qPCR) in whole brain and manually dissected brain regions of adult *Snord115*-deficient mice and WT littermates, respectively (n = 6 per genotype). RNA2-to-RNA1 ratios in WT were set to 1. Right: Histograms show the RNA2-to-RNA1 ratio (RT-qPCR) in hypothalamus and choroid plexus of adult WT mice, respectively (n = 5–6 per tissue). The RNA2-to-RNA1 ratio in hypothalamus was set to 1 (**D**) Left: Ethidium bromide staining of RT-PCR products obtained in hypothalamus of *Snord115*-deficient and WT littermates using a pair of primers (horizontal black arrows) that simultaneously detect RNA2 and RNA1, as illustrated above the gel. M, Marker (bp). Right: intensity of PCR signals was quantified and RNA2-to-RNA1 in WT was set to 1.

different HTR2C receptors translated from 32 distinct mRNAs. Remarkably, RNA editing influences HTR2C response with increased RNA editing correlating with diminished G-protein coupling efficacy and reduced constitutive activity. Of note, editing at the C- and E-sites plays a pivotal role in modulating the HTR2C signaling pathway (*Berg et al., 2001*; *Burns et al., 1997*; *Fitzgerald et al., 1999*; *Niswender et al., 1999*; *Wang et al., 2000*). Given that introducing a 2'-O-methylation at an editing site specifically decreases ADAR-mediated deamination in vitro (*Yi-Brunozzi et al., 1999*), we logically anticipated that editing at the target C-site should increase in *Snord115*-deficient mice as previously proposed based on artificial, non-neuronal cell systems (*Vitali et al., 2005*). To address this, we quantified A-to-I RNA editing in whole brain and spinal cord as well as in eight manually dissected brain regions using deep sequencing of cDNA of spliced *Htr2c* mRNA followed by scoring A-to-G mismatches between RNA-seq reads and genomic sequence (*Figure 3A* -lower

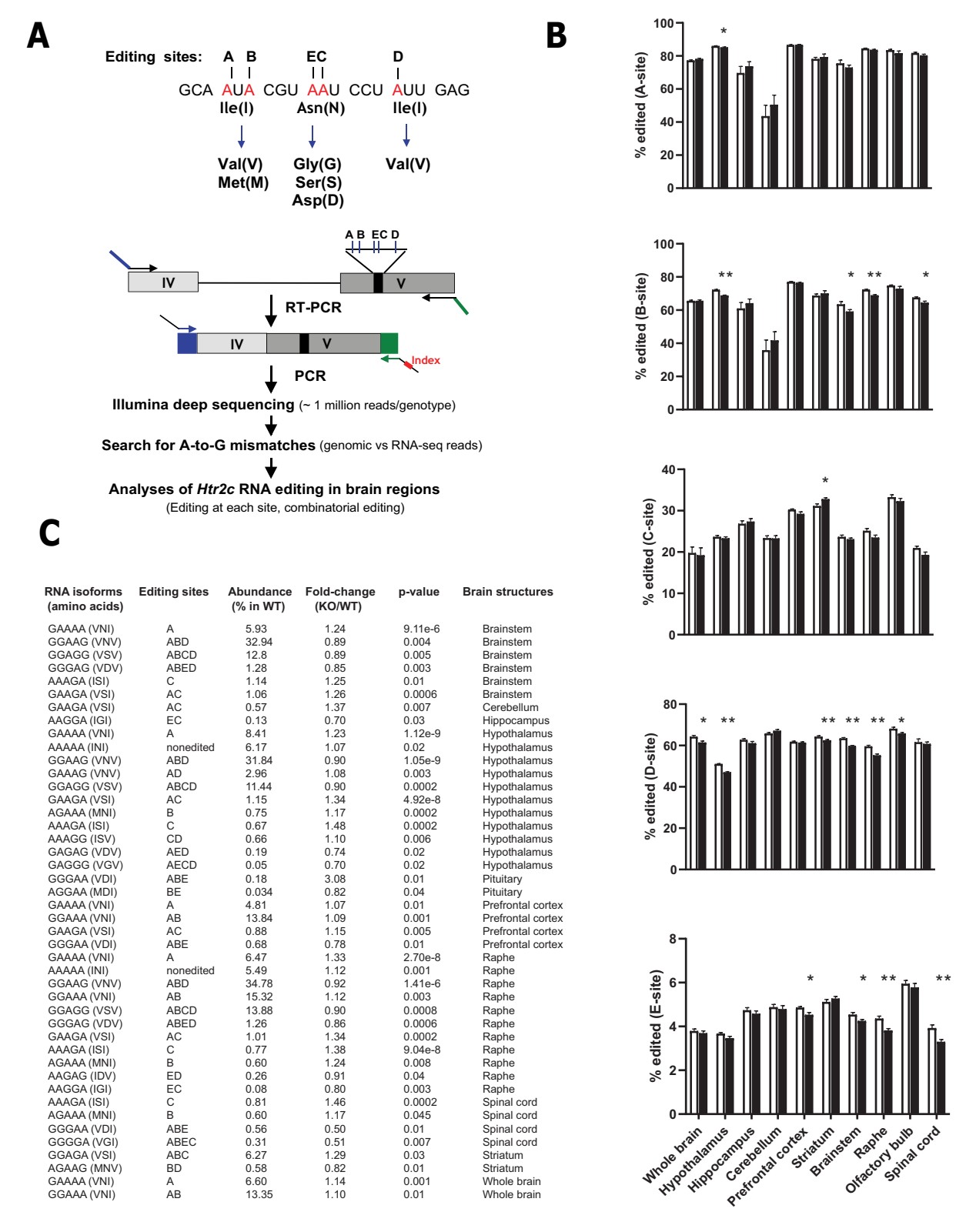

**Figure 3.** Loss of *Snord115* has a mild influence on *Htr2c* RNA editing profiles. (**A**) Upper part: sequence-specific A-to-I RNA editing alters the genetic information embedded in exon V of *Htr2c* mRNA. Lower part: High-throughput sequencing method to measure RNA editing of *Htr2c* mRNA. (**B**) Percentage of RNA editing at each individual site in whole brain, spinal cord as well as in eight manually dissected brain regions of *Snord115*-deficient (black bars) mice and WT littermates (white bars; n = 9–12 per genotype). (**C**) The table lists significant changes (p<0.05) in the proportion of *Htr2c*
*Figure 3 continued on next page*

Figure 3 continued

isoforms in *Snord115*-deficient brain regions as indicated by the KO-to-WT ratio. The relative abundance of differentially expressed *Htr2c* isoforms, as measured in WT, is also indicated (%).

The online version of this article includes the following figure supplement(s) for figure 3:

**Figure supplement 1.** Loss of Snord115 and/or Snord116 has a mild impact on Htr2c RNA editing during development or after chronic mCPP treatment.

part). This high-throughput sequencing method allows reliable analyses of RNA editing (*Morabito et al., 2010b*). Indeed, given the high sequencing coverage (~1 million reads per genotype) and the substantial number of mice analyzed (n = 8–12 per genotype), there was little inter-mouse variability, with highly similar editing patterns to those previously published: the A, B, and D sites being more edited compared to C and E sites (*Abbas et al., 2010*; *Morabito et al., 2010b*). Our analyses showed that editing frequencies at each of the five individual sites remained globally unaltered in *Snord115*-KO mice. Nevertheless, we identified a few alterations at a magnitude ranging from 1 to 4%, particularly in the hypothalamus, brainstem, and raphe (*Figure 3B*). Regarding the relative proportions of *Htr2c* isoforms (*Figure 3C*), we observed altered expression for 44 isoforms, with most of them (28 out of 44) corresponding to lowly expressed *Htr2c* transcripts accounting for less than 5% of the total. These changes could not have been anticipated in a straightforward manner based on the most parsimonious interpretation stating that *Snord115* targets the C-site for 2'-O-methylation (*Cavaillé et al., 2000*; *Vitali et al., 2005*). Because *Htr2c* RNA editing increases over time, from embryogenesis to adulthood (*Wahlstedt et al., 2009*), we reasoned that lack of *Snord115* could be more detrimental during dynamic changes in RNA editing. We thus quantified editing at E16.5 of development in whole brain as well as in hypothalamus, cortex, and hippocampus. As expected, adult tissues displayed an overall increase in editing as compared to their E16.5 counterparts. However, the comparison of *Snord115*-deficient samples with WT did not reveal any significant differences in editing levels at any of the five sites, except for slight increase at the B-site and a very few changes in mRNA isoforms for prefrontal cortex (*Figure 3—figure supplement 1*). Finally, *Htr2c* RNA editing profiles also remained largely unaltered in hypothalamus of adult mice chronically treated with the preferential HTR2C agonist meta-chlorophenylpiperazine (mCPP), or even in P1 or adult hypothalamus of mice simultaneously deleted for *Snord115* and *Snord116* genes (*Figure 3—figure supplement 1*). Altogether, our high-throughput sequencing strategy found little support for a pronounced effect of *Snord115* on the regulation of *Htr2c* RNA editing in vivo and the functional relevance, if any, of the observed small magnitude changes remain uncertain.

## *Snord115*-deficient mice display normal emotional responses and sociability

We then sought to determine whether deleting the brain-specific *Snord115* genes is associated with behavioral and/or metabolic abnormalities. Because altered HTR2C signaling, including abnormal RNA editing, is linked to anxiety disorders and depressive states (*Chagraoui et al., 2016*; *Heisler et al., 2007*; *Mombereau et al., 2010*; *Spoida et al., 2014*), we first subjected a cohort of 8–12 week-old *Snord115*-deficient and WT mice to the open field (OF) test. This classical behavioral test simultaneously measures novelty-induced exploration and certain components of anxiety. As judged by the distance travelled (*Figure 4A* -left), the exploratory drive and locomotion were not altered in *Snord115* mutant mice. Anxiety levels of *Snord115*-KO mice were also in the normal range, as evidenced by the number of entries (not shown) and time spent (*Figure 4A* -right) in the central zone of the OF considered as anxiogenic for rodents. The normal status of anxiety was further confirmed through the elevated plus maze (EPM) test, a cross-shaped platform that consists of two open and two closed arms elevated from the floor. Here, anxiety-like behavior is reflected by avoidance of the two open (anxiogenic) arms. Again, the frequencies of entry (not shown) and time spent (*Figure 4B*) in the open arms of *Snord115*-deficient mice were found indistinguishable from that of their WT littermates.

To evaluate other forms of emotional response, the novelty-suppressed feeding (NSF) test was used. In this test, overnight food-deprived mice are exposed to a novel, illuminated environment in which a pellet of food is presented in the center. Under these conditions, any delay in food

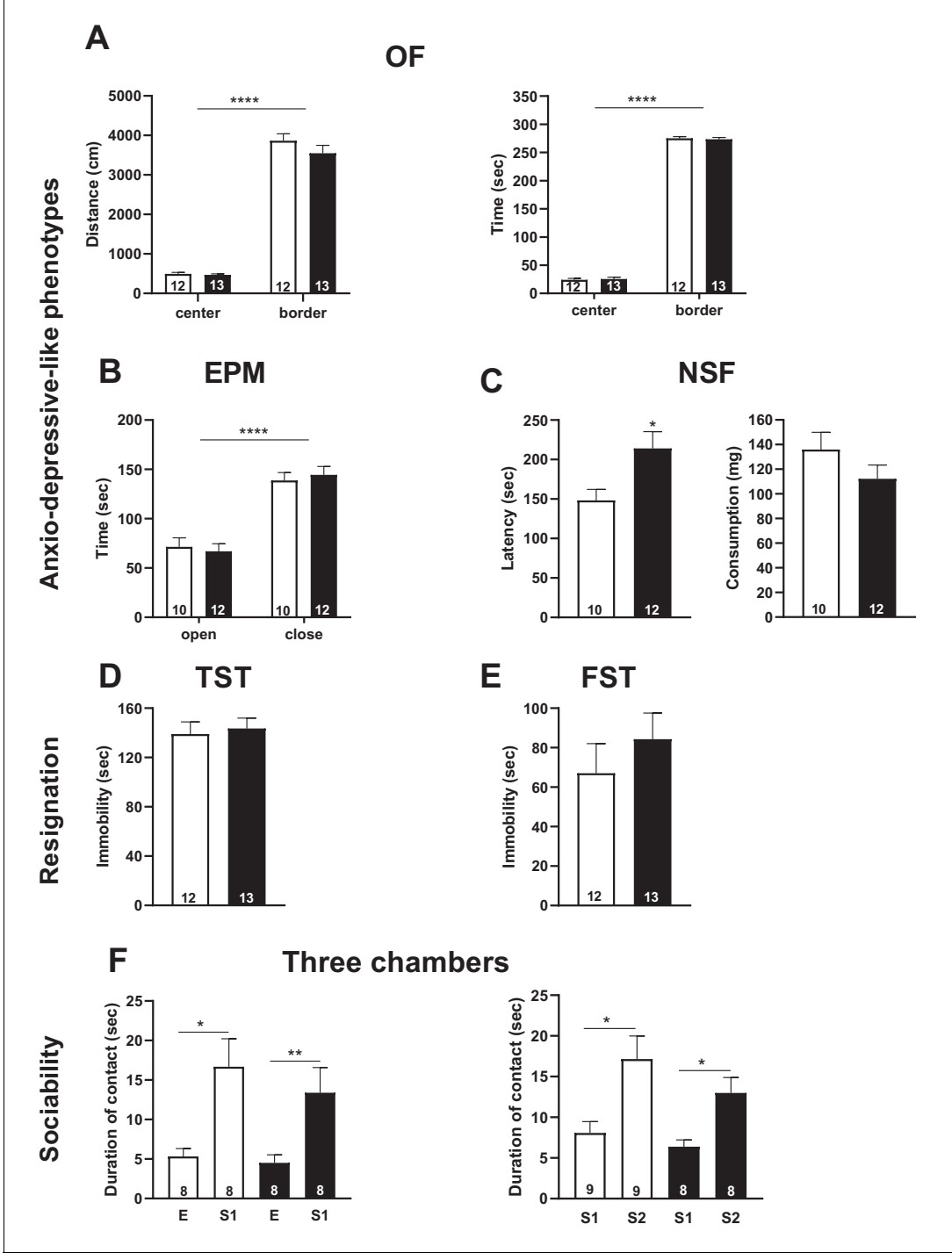

**Figure 4.** *Snord115*-deficient mice do not display abnormal anxio-depressive-like phenotypes and show normal response to sociability stimuli. Cohorts of adult *Snord115*-deficient males (black bars) and their WT littermates (white bars) were subjected to Open field (A), Elevated plus maze (B), Novelty supressed feeding (C), Tail suspension (D), Forced swimming (E) and Three chambers (F) tests. The number of individuals analyzed are indicated within histograms. Note that due to tracking issues, one WT individual was omitted from analysis (4F-left-panel). None of these analyses showed a significant effect of genotype (ANOVA p-values>0.05 for genotype in each panel).

consumption is interpreted as mixed anxio-depressive-like behaviors. Interestingly, *Snord115*-KO mice showed an increase in the latency to feed compared to WT (*Figure 4C* -left). These results cannot be attributable to change in hunger since food consumption in the home cage 5 min after the test showed no differences between both groups (*Figure 4C* -right). This prompted us to employ two additional tests - the tail suspension (TST) and forced swim (FST) tests - that also interrogate a core symptom of depression such as resignation. In TST, mice are suspended by their tails with impossibility to hold onto nearby surfaces while in FST they are placed in an inescapable tank filled with water. In both paradigms, the escape behavior is monitored with immobility time used as a read-out of despair-like phenotype. As shown in *Figure 4D and E*, the duration of immobility of *Snord115*-deficient mice were similar to that of WT. Collectively, even though altered behavior was apparent in NSF, we nevertheless consider that *Snord115*-deficient mice are able to mount a normal emotional response when exposed to various acute stressors.

Mice expressing the fully edited (VGV), less active HTR2C receptor display elevated levels of social interaction including increased aggressive behaviors (*Martin et al., 2013*). We then investigated the sociability of *Snord115*-deficient mice through the three-chamber test. During the habituation session for which there is no additional mouse in the cages, WT and mutant mice contacted equally the two empty (E) compartments, thus excluding any bias for one of the two compartments (not shown). The day after, a non-familiar mouse (S1) was introduced in one of the two compartments and time spent interacting with S1 over E1 was used as an index of sociability. As shown in *Figure 4F* -left, both WT and *Snord115*-deficient mice spent more time in contacting S1 with no difference between the two genotypes. Of note, we did not detect any apparent signs of aggression. Finally, we also tested the preference for social novelty by introducing another stranger (S2) mouse. Again, we did not find any differences between WT and KO mice with, as expected, both genotypes interacting more with the novel intruder (S2) than the familiar one (S1) (*Figure 4F* -right). Taken together, these observations showed that social preference and recognition are unchanged in *Snord115*-KO mice.

## *Snord115*-deficient mice show normal spontaneous food intake and energy balance on chow diet

*Snord115* is expressed in most, if not all, pro-opiomelanocortin (*Pomc*) neurons in the arcuate nucleus (ARC) of the hypothalamus (*supplementary file 2*). The activation of HTR2C in ARC *Pomc* neurons, but also in nucleus of the solitary tract, promotes satiety and it also regulates body weight, energy balance, and glucose homeostasis (*Berglund et al., 2013*; *D'Agostino et al., 2018*; *Heisler et al., 2002*; *Xu et al., 2010*; *Xu et al., 2008*; *Zhou et al., 2007*). We then examined feeding behaviors and energy balance. We found that body weights (*Figure 5A*) and body composition (*Figure 5B*) of 8-week-old adult *Snord115*-KO mice were in the normal range as compared to WT littermates. Moreover, blood glucose levels were also unaffected and, after a bolus administration of glucose, WT and *Snord115*-deficient mice displayed the same ability to regulate a glucose load (*Figure 5C*). Food intake of *Snord115*-KO mice was then automatically quantified over a 24 hr period using the TSE Phenomaster system. Both daily and nightly food intake (*Figure 5D*), as well as physical activity (*Figure 5E*), remain unaffected in mutant mice while, as expected, increased food consumption and elevated locomotion during the dark phases. Finally, indirect calorimetry did not detect any defect in energy expenditure (*Figure 5F*) or respiratory exchange ratio (RER; *Figure 5G*), either during daylight or at night. Thus, feeding behavior and energy metabolism are unlikely to be disrupted by loss of *Snord115*.

## *Snord115*-deficient mice display normal response to starvation

The normal daily food intake of *Snord115*-deficient mice when nutrients are constantly available does not rule out potential deficiencies upon food deprivation. We then assessed response of *Snord115*-KO mice after 18 hr (overnight) fasting. This was first done by measuring the steady-state levels of three ARC mRNAs encoding appetite-associated neuropeptides whose expression is affected by fasting (*Baldini and Phelan, 2019*; *Henry et al., 2015*). *Figure 5H* shows that our starvation procedure led, as expected, to an increase in the hypothalamic expression of the orexigenic *Npy* and *Agrp* peptide encoding mRNAs in WT mice while expression of the anorexigenic *Pomc* peptide encoding mRNA exhibited a tendency to decrease. Remarkably, the three above-mentioned

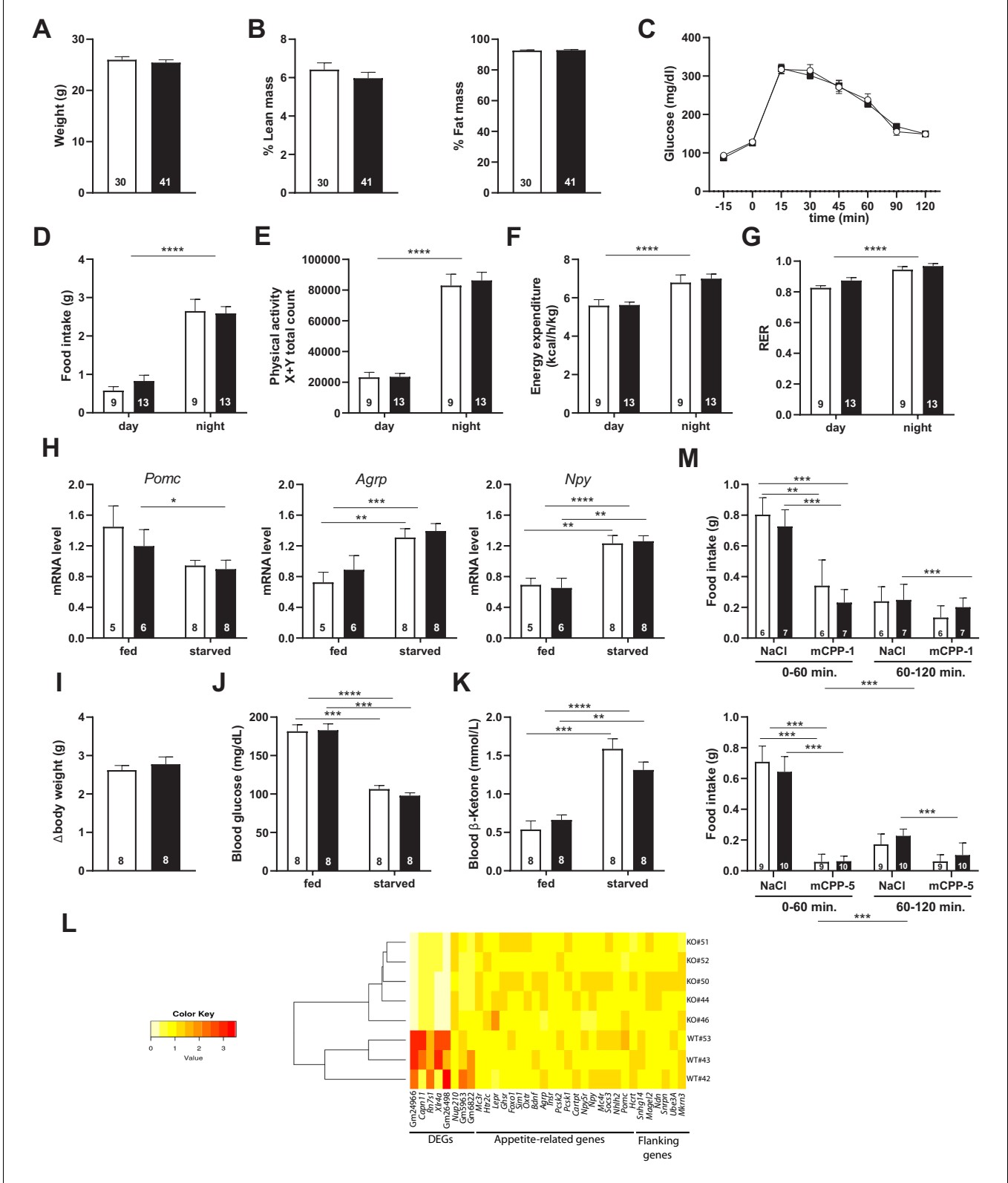

**Figure 5.** *Snord115*-deficient mice show regular homeotic feeding behavior and energy balance. Body weight (**A**) and body composition measured by EchoMRI (**B**). Glucose tolerance test (GTT) of overnight (16 hr) fasted *Snord115*-deficient mice (n=14) and their WT littermates (n=9) (black and white squares, respectively). The graph represents the fate of glucose versus time after i.p. glucose administration (**C**). Food intake (**D**), spontaneous locomotion (**E**), energy expenditure (**F**), and respiratory exchange ratio (**G**) of *Snord115*-deficient mice and WT littermates were measured using the TSE

*Figure 5 continued on next page*

*Figure 5 continued*

PhenoMaster System. (H-K): *Snord115*-deficient mice and WT littermates were overnight (18 hr) starved. Expression levels of *Pomc, Agrp* and *Npy* mRNAs in hypothalamus were measured (RT-qPCR relative to *Gapdh*) (H). Weight loss after fasting (I), blood glucose (J) and ketone bodies (K) levels were measured in the fed and starved states. (L) Heatmap of the normalized mRNA-seq read counts from a list of genes of interest, as indicated below the panel. The normalized expression for each gene (column) and each genotype (row) is represented by colour code (yellow, lower value; red, higher value). Note that Gm24966 and Gm26498 match the deleted region. (M) *Snord115*-deficient mice and WT littermates were overnight (16 hr) starved and food intake of mCPP- and NaCl-treated mice was recorded using the TSE Phenomaster system during the first and second hour after treatment. mCPP was i.p. administered (1 mg/kg, M-top) or (5 mg/kg, M-bottom). White and black bars represent WT and *Snord115*-deficient mice, respectively. The number of individuals analysed are indicated within histograms. None of the analyses in panels A-K and M showed a significant effect of genotype (ANOVA p-values>0.05 for genotype in each panel).

mRNAs were roughly similarly regulated in *Snord115*-deficient mice during fasting. The overall physiological response to fasting was also unaffected, as evidenced by body weight loss (*Figure 5I*), decreased glucose levels (*Figure 5J*) and increased ketone level (*Figure 5K*). Finally, mRNA-seq also showed that, in the absence of *Snord115*, global gene expression in hypothalamus of mice fed on regular chow diet is unaffected compared to WT. Indeed, we only uncovered eight differentially expressed transcripts (DEG) with two of them matching the deleted genomic region (*supplementary file 3*). Hence, the expression of several key appetite-related genes, as well as that of the neighbouring imprinted genes, was in the normal range (*Figure 5L*). These data suggest that *Snord115*-deficient mice respond normally to starvation and do not present any apparent defects in hypothalamic gene circuits of the melanocortin pathway.

## *Snord115*-deficient mice respond normally to the appetite-suppressant actions of the HTR2C agonist

HTR2C is the main, if not the sole, target on which the preferential HTR2C agonist mCPP exerts its appetite-suppressant effects (*Berglund et al., 2013*; *Tecott et al., 1995*). We then reasoned that *Snord115*-deficient mice should display an attenuated mCPP response, that is a decreased sensitivity to the actions of mCPP. Vehicle (saline solution) or mCPP were administered intra-peritoneally to 16 hr fasted WT and KO mice. Chow diet was then reintroduced 30 min later and the amount of food consumed was recorded using the TSE Phenomaster system during the first and second hour following food presentation. *Figure 5M* (upper and lower panels) shows that food intake of NaCl-treated *Snord115*-deficient mice was similar than that of WT littermates, indicating that homeostatic feeding is normal. This was consistent with unaffected neural circuits of the melanocortin pathway as described above. As expected, food intake of mCPP-treated WT and KO mice was severely and dose-dependent reduced during the first hour as compared to controls (50% and 90% decrease following administration of 1 and 5 mg/kg of mCPP, respectively). Given that anorectic responses of mCPP remained in the same range in both genotypes, loss of *Snord115* does not perceptibly disturb the hypothalamic HTR2C signaling cascade in vivo, at least when fast-induced food intake is used as a read-out.

## *Snord115*-deficient mice behave normally upon high-fat diet

We next asked whether metabolic stressors, such as long-term exposure to high-fat diet (HFD), may reveal deficiencies in feeding behavior and/or energy homeostasis not seen on regular chow diet. A cohort of 8- to 12 week-old adult mutant and WT mice were then subjected to HFD (60% lipid) for 24 weeks. During the course of diet-induced obesity, body weight gain of *Snord115*-KO mice was similar to WT mice (*Figure 6A*) with normal adiposity as evidenced by the weights of perigonadal, subcutaneous fat mass and liver measured after euthanasia. Only brown adipose tissue showed a statistically significant change, with *Snord115*-KO tissues being 38% heavier than WT tissues (*Figure 6B*). We then searched for potential differences in glucose metabolism by performing an oral glucose test tolerance after 12 weeks of HFD exposure. Capability for glucose clearance in *Snord115*-KO mice did not differ substantially from that of WT littermates (*Figure 6C*). In addition, indirect calorimetry analyses did not reveal any obvious differences in food intake (*Figure 6D*), ambulatory activities (*Figure 6E*), energy expenditure (*Figure 6F*) or respiratory exchange ratio (*Figure 6G*). Altogether, our analyses did not unveil significant metabolic differences between *Snord115*-KO and WT littermates after HFD exposure.

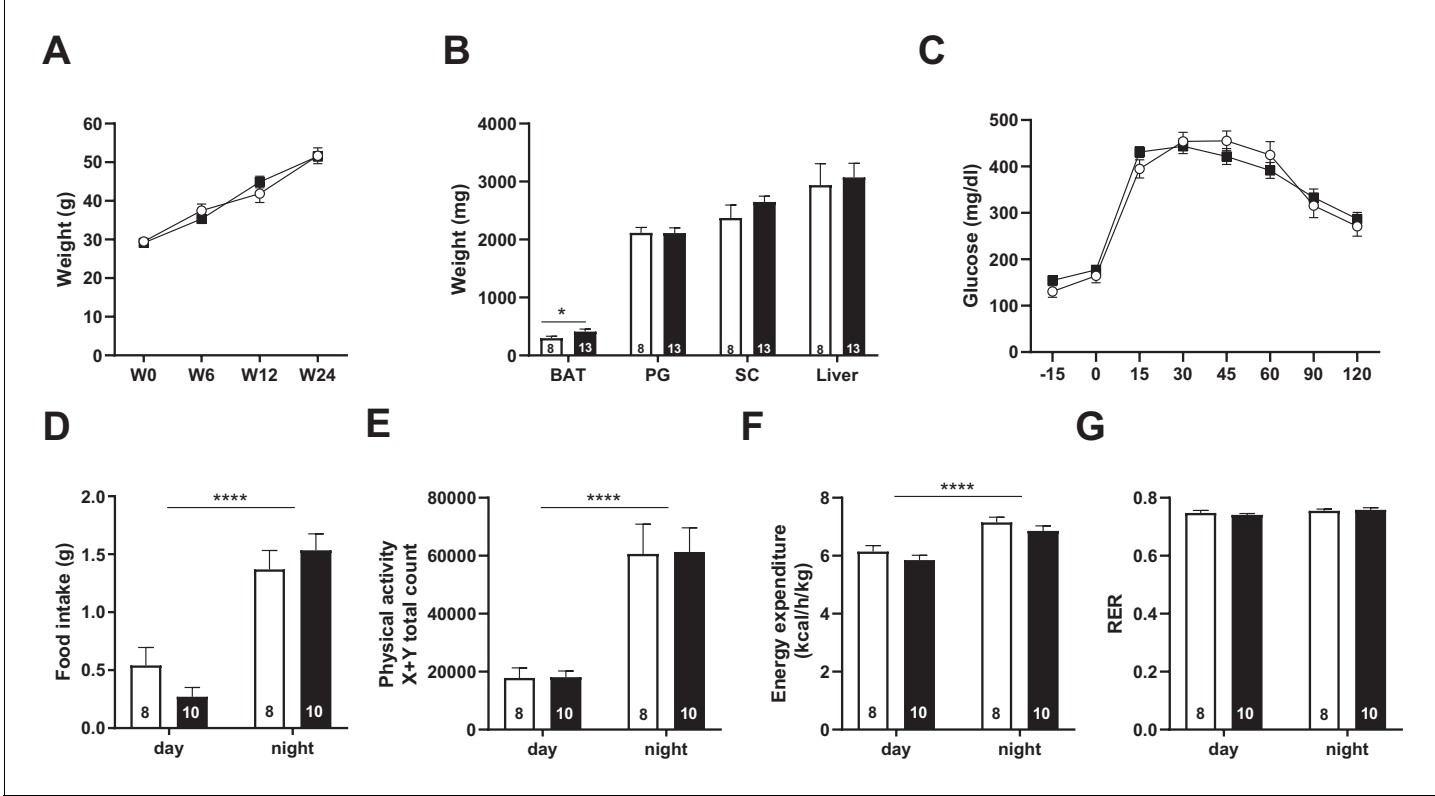

**Figure 6.** *Snord115*-deficient mice behave normally when fed a high-fat diet. (**A**) Weight of *Snord115*-deficient mice and WT littermates before (W0) and after 6 (W6), 12 (W12) and 24 (W24) weeks of HFD. (**B**) Adiposity of *Snord115*-deficient mice and WT littermates after 24 weeks of HF diet as judged by the weights of brown adipose tissue (BAT), perigonadal (PG) fat, subcutaneous (SC) fat and liver. (**C**) Glucose tolerance test of overnight (16 hr) fasted *Snord115*-deficient (n=13) and WT littermates (n=8) after 12 weeks of HFD. The graph represents the fate of glucose versus time after oral glucose administration (1.5 g/kg). Food intake (**D**), spontaneous locomotion (**E**), energy expenditure (**F**) and respiratory exchange ratio (**G**) of *Snord115*-deficient mice and WT littermates were measured after 16 weeks of HFD (Phenomaster TSE System). White and black bars represent WT and *Snord115*-deficient mice, respectively. The number of individuals analyzed are indicated within histograms. None of these analyses showed a significant effect of genotype (ANOVA p-values>0.05 for genotype in each panel), except for panel B (p-value=0.04125 for genotype, with pairwise t-tests indicating a significant difference between WT and KO brown adipose tissues).

## Discussion

In this study, we challenged long-standing observations linking the neuronal-specific *SNORD115* genes to activity of the HTR2C receptor and clinical features of PWS. More precisely, we aimed at testing, for the first time and at the organism level, the hypothesis according to which *SNORD115* regulates the processing of *HTR2C* pre-mRNA and, in doing so, influences HTR2C-mediated brain functions. Of particular translational relevance, lack of *SNORD115*, and thus putative defects in post-transcriptional regulation of *HTR2C* pre-mRNA could, in principle, account for impaired emotional response and/or food seeking behavior that feature PWS (*Cavaillé, 2017*; *Marty and Cavaillé, 2019*; *Stamm et al., 2017*).

Despite the fact that we thoroughly assessed post-transcriptional regulation of *Htr2c* pre-mRNA in numerous discrete brain regions collected from a substantial number of genetically comparable WT and *Snord115*-KO mice, we failed to detect any discernible impact on alternative splicing, as judged by the regular-to-truncated ratio of alternatively spliced *Htr2c* mRNA isoforms. Furthermore, only subtle changes in regional-specific RNA editing of functional relevance, if any, were uncovered. Our observations are in line with two recent in vivo studies. Reactivation of the *Snord116-Ipw-Snord115* genomic interval in choroid plexus - that does not normally express *Snord115* - leads to modest impact on RNA editing without any impairment in alternative splicing of *Htr2c* pre-mRNA (*Raabe et al., 2019*). Moreover, oligonucleotides mimicking *Snord115* base-pairing have no effects on alternative splicing, despite the fact that in vivo delivery of antisense oligonucleotides targeting

other RNA segment of *Htr2c* pre-mRNA can alter the regular-to-truncated ratio of HTR2C (*Zhang et al., 2016*). Hence, in vivo molecular outputs of SnoRD115/*Htr2c* mRNA interaction are more complex than previously inferred from in vitro observations (*Kishore and Stamm, 2006*; *Vitali et al., 2005*).

Our findings therefore may appear at odds with prior observations, including our own (*Cavaillé et al., 2000*; *Vitali et al., 2005*). These apparent discrepancies deserve to be tempered by considering them in their full scientific context. First, in vitro studies with non-neuronal cells overexpressing both *Htr2c* and *Snord115* from artificial mini-genes do not take into account differential cellular compartmentalization of the two interacting RNA partners (*Kishore and Stamm, 2006*; *Vitali et al., 2005*). In addition, it has been shown that 'any cellular RNA' can be modified, at least to some extent, by forced expression of an exogenously expressed SNORD (*Cavaillé et al., 1996*; *Kiss-László et al., 1996*). Therefore, in vitro data must be interpreted with caution and, as such, they must be considered as proof-of-principles but by no means as definitive functional proofs. Second, conflicting observations regarding the precise mode of action of *Snord115* still remained. Changes in *Htr2c* alternative splicing were described in the hypothalamus of PWS-IC mice (*Garfield et al., 2016*) but not in those of PWS patients studied (*Bochukova et al., 2018*) or in whole brain of PWS-IC mice (*Doe et al., 2009*). Third, although altered *Htr2c* RNA editing was reported in the brain of PWS-IC and PatDp/+ mouse models, these changes did not correlate with the dosage of *Snord115* (*Doe et al., 2009*; *Nakatani et al., 2009*). In addition, no reproducible editing changes were noticed in the frontal cortex of a few PWS patients (*Glatt-Deeley et al., 2010*). Analyses of RNA editing in these studies involved a very small number of sequenced *HTR2C* cDNA clones, typically ~20–100 per genotype. Given that the combinatorial RNA editing of the *HTR2C* pre-mRNA produces 32 mRNA edited isoforms, low-coverage sequencing procedures are subject to strong sampling biases and low statistical power, thus rendering the interpretation of changes in the level of edited *HTR2C* isoforms highly risky. As already correctly pointed out by others (*Abbas et al., 2010*; *Morabito et al., 2010b*; *O'Neil and Emeson, 2012*), recent high-throughput sequencing methods, as described here with ~1 million sequences per genotype, have made it feasible to resolve such technical issues. Accordingly, our findings are much more statistically robust than previous published studies. In such a complex framework, demonstrating the presence of a 2'-O-methyl group at the C-site would undoubtedly represent ultimate proofs to define the mode of action of *Snord115*. However, identifying a ribose methylated nucleotide in low abundance *Htr2c* transcripts remains technically challenging, in particular if methylated at sub-stoichiometric levels.

The question then arises as to whether loss of *Snord115* is accompanied by physiological abnormalities. Eating disturbance and obesity in PWS (*Cassidy et al., 2012*), together with the anorectic proprieties of HTR2C (*Berglund et al., 2013*; *D'Agostino et al., 2018*; *Heisler et al., 2002*; *Xu et al., 2010*; *Xu et al., 2008*; *Zhou et al., 2007*), prompted us to first concentrate on food intake and energy balance. Our genetic background and experimental settings, however, failed to reveal any notable alterations on regular diet, upon fasting-refeeding transition or even after exposure to HFD. Of note, blunted feeding effects of the HTR2C agonist (WAY-161503) and impaired melanocortin signaling were described in the PWS-IC mice. These defects were attributed to lack of *Snord115* (*Garfield et al., 2016*). The PWS-IC mouse model carries a paternally inherited deletion of the imprinted center (IC) region leading to the silencing of all imprinted genes positioned on the paternal allele (*Yang et al., 1998*). Moreover, these knockout mice are growth retarded and it is necessary to switch to a hybrid genetic background to avoid 80–100% neonatal mortality due to absence of paternal contribution. These inherent limitations represent obvious sources of confounding factors. Thus, we favor an alternative explanation according to which defects in HTR2C-mediated feeding circuitries result from a combination of the deficiencies of several paternally expressed genes. This is all the more likely as defects in the serotonergic system have been described in mice disrupted for the paternally expressed *Magel2* (*Mercer et al., 2009*) or *Ndn* (*Zanella et al., 2008*) genes. Moreover, altered expression of neuropeptide of the melanocortin system was also reported in *Snord116*-KO mice (*Burnett et al., 2017*; *Qi et al., 2016*). Thus, *Snord115* appears largely dispensable for homeostatic feeding and, as such, it considerably differs from *Snord116* ablation which leads to decreased body-weight, apparent increased food intake and resistance to HFD-induced obesity, among other phenotypes (*Ding et al., 2008*; *Polex-Wolf et al., 2018*; *Qi et al., 2016*; *Skryabin et al., 2007*; reviewed in *Marty and Cavaillé, 2019*).

The apparent lack of profound molecular and behavioral consequences does not necessarily mean that *Snord115* is of minor relevance. As a case in point, we observed changes in the excitability of 5-HT neuronal circuitries in living *Snord115*-KO mice (B. Guiard, personal communication). More broadly, the lack of *SNORD115* might only have observable phenotypic consequences in sensitized genetic contexts, for example when expression of the other flanking co-regulated paternally expressed genes is also disrupted as observed in the vast majority of PWS people or in PWS-IC mice. Consistent with that possibility, mice simultaneously deleted for *Snord115* and *Snord116* genes display more severe phenotypes than single *Snord115*- or *Snord116*-KO mice (Boursereau, Marty et al. unpublished observations). Finally, and in the same vein, loss of *SNORD115* might only become apparent under perturbed metabolic contexts, perhaps as 'secondary disease-effects' once obesity is fully established in individuals with PWS.

There are other limitations commonly known to blur phenotypes in constitutive KO models. An obvious caveat is that any tissues, and especially brain, are composed of heterogeneous cell populations. For instance, transcriptomic analyses of single cells in mouse hypothalamus identified 34 neuronal cell clusters (*Chen et al., 2017*). *Pomc* neurons represent a few per cent of these hypothalamic neurons with only 30–40% expressing *Htr2c* (*Lam et al., 2017*). Assuming that *Snord115* mostly regulates *Htr2c* processing in the minority of *Pomc* neurons, it might then be illusory to hope detecting robust molecular changes in a mixture of molecularly and phenotypically diverse neurons. An additional layer of complexity may also arise from poorly understood 'adaptive mechanisms' which increase the density of the fully edited (VGV) HTR2C receptor at the cell membranes, very likely compensating for its diminished activity (*Kawahara et al., 2008*; *Morabito et al., 2010a*; *Olaghere da Silva et al., 2010*). Finally, how site-specific RNA editing is regulated remains a topic of intense research (*Freund et al., 2020* and references therein). Initial studies based on Sanger sequencing methods claimed that *Htr2c* editing is dynamically regulated in a context-dependent manner, particularly depending on serotonin levels (*Schmauss, 2003*). Yet, more recent high-throughput sequencing analyses did not fully confirm all these earlier findings. Only rather subtle RNA editing changes were noticed with respect to genetic background or serotonin content (*Abbas et al., 2010*; *Morabito et al., 2010b*), disease states (*O'Neil and Emeson, 2012*; *Zhu et al., 2012*), pharmacological manipulations (*Abbas et al., 2010*) or reovirus infections (*Hood et al., 2014*). Hence, *Htr2c* RNA editing appears robustly controlled very likely by distinct overlapping mechanisms and *Snord115* might only be one of these regulators. Keeping these potential limitations in mind, future studies involving conditional *Snord115* gene deletion together with single neuron analyses are now required to fully apprehend how loss of *SNORD115* may shape HTR2C-mediated neuronal functions and thus possibly the pathophysiology of PWS.

## Materials and methods

### Mice housing and breeding

Animal procedures were approved by the University of Toulouse and CNRS Institutional Animal Care Committee (DAP2016061716367988 and DAP2018011214542827). The animal housing facility met CNRS standards. Mice were housed in a temperature-controlled room with a 12 hr light–dark cycler and access to water and food ad libitum, either regular chow diet or HFD diet (ssniff DIO – 60 kJ% fat [Lard]). We backcrossed to C57BL/6J genetic background for at least eight generations before proceeding to behavioral and metabolic analyses. Upon euthanasia, tissues were harvested, weighted and immediately snapfrozen in liquid nitrogen and stored at −80˚C.

### Generation of *Snord115* knock-out mouse model

A *Snord115* knock-out mouse model was generated by co-injecting zygotes isolated from (C57BL/6JxCBA) F1 mice with in vitro transcribed Cas9 encoding mRNA (GeneArt CRISPR nuclease mRNA, Invitrogen) and two guide RNAs (MEGAshortscript T7 Transcription Kit; Ambion) designed to basepair upstream and downstream of *Snord115* gene array (crispr.mit.edu tool). Briefly, prepubertale (C57BL/6JxCBA) F1 female (4–6 weeks old) were superovulated by IP injection of 5IU PMS and 5IU hCG at an interval of 46 hr and mated overnight with (C57BL/6JxCBA) F1 adult male mice. Zygotes were collected after 22 hr of hCG injection. Microinjection was performed using an Eppendorf microinjector. Typically, a few picoliters of Cas9 mRNA (100 ng/µL) and in vitro transcribed sgRNAs (25

ng/µL each) were co-injected into cytoplasm and pronuclei of zygote. Injected zygotes were then cultured in M16 medium until blastocyst stage for analyzing cleavage efficiency or directly transferred into the oviductal ampullas of pseudopregnant female mice (around 20 embryos per female). Following the CRISPR/Cas9 procedure, pups born were selected by PCR on genomic DNA from tail tips (Wizard Genomic DNA Purification Kit; Promega). Primers are listed in *Supplementary file 4*.

## RNA extraction, RT-qPCR, and Northern blot

Total RNA was extracted using TRI reagent (Euromedex) followed by RNAse free-RQ1 DNAse (Promega) and proteinase K (Sigma) treatments. 1 µg of whole cell RNA was reverse transcribed with random hexamer primers using Go Script Reverse Transcriptase kit (Promega) at 42℃ for 60 min. mRNA expression was performed using the IQ Custom SYBR Green Supermix (Bio-Rad) qPCR. The relative quantification of gene expression was performed using the standard curve method with triplicates for each data point. For Northern blotting analysis, 5 µg of total RNA was fractionated by electrophoresis on a 6% polyacrylamide/7 M urea denaturing gel. Electro transferred onto a nylon membrane (Amersham Hybond-N, GE Healthcare) followed by UV crosslinking (Stratalinker). Hybridizations were carried out with 5′-end $^{32}$P-labeled-DNA oligonucleotide probes. Membranes were incubated overnight at 50℃ in 5X SSPE, 5X Denhardt's, 1% SDS, 150 µg/mL yeast tRNA and washed twice in 0.1% SSPE, 0.1% SDS for 15 min at room temperature. a Typhoon Biomolecular Imager (Amersham) and visualized using Multi Gauge V3.0 software. Radioactive signals were revealed using a Typhoon Biomolecular Imager (Amersham) and visualized using Multi Gauge V3.0 software. Primers are listed in *Supplementary file 4*.

## A-to-I RNA editing analyses

A 249-nt long segment overlapping the spliced Exon IV-Exon V junction of *Htr2c* mRNA was amplified with *Htr2c*-F and *Htr2c*-R primers (35 amplification cycles; annealing temperature of 65℃). Single multiplexing was performed using home-made 6 bp index which were added to reverse primer during a second PCR (12 cycles) using forward P5 and reverse P7 primers. Primers are listed in *Supplementary file 4*. The resulting PCR products were purified and sequenced on a HiSeq3000 (Illumina) according to the manufacturer instructions. The quality of the run was checked internally using PhiX (15%) and each paired-end sequence was assigned to its sample with the help of the previously integrated index. Reads were aligned on the mm10 mouse genome using hisat2 (*Kim et al., 2019*) and reads mapping on the amplicon locus were selected using bedtools. For each of the five editing sites, A-to-G conversion was scored in every read with high quality (sequencing quality score ≥20 for the nucleotide mapped on that site). Raw data are available on Sequence Read Archive (SRA) database under the accession numbers PRJNA603261 and PRJNA603264. Scripts used for that analysis, detailed instructions and intermediary data have been deposited at https://github.com/HKeyHKey/Hebras_et_al_2020 (*Seitz, 2020*; copy archived at swh:1:rev:1b96b6f5e8d479eb43f12f6687f04d4a60f4e305).

## rRNA 2′O-methylation profiling

RiboMeth-seq was performed in biological triplicates on whole adult mouse brains (*Birkedal et al., 2015*). In brief, 5 µg of total RNA was alkaline degraded and size fractionated to the range of 20–40 nt. Sequencing adaptors were ligated on to the ends of the alkaline degraded RNA using a modified tRNA ligase. The library RNA containing adaptors was reverse transcribed using Superscript IV (Thermo Fisher Scientific). The final libraries were sequenced on a PI Chip v3 (Thermo Fisher Scientific) using an Ion Proton semiconductor sequencer. Reads were mapped against curated mouse rRNA sequences (*Hebras et al., 2020*). RiboMeth-seq data are available on GEO under the accession number GSE145159.

## Transcriptomic analyses

Library preparation (polyA selection) and Illumina sequencing (2 × 150 bp configuration) were performed by Genewiz. Raw data quality control was done using FASTQC (Version 0.11.9). Reads were aligned to reference genome (GRCm38/mm10, Dec. 2011) with STAR (Version 2.4.0.1). Read counts were generated for each annotated gene (GRCm38/mm10, Ensembl gtf release 91) using HTSeq-count (Version 0.11.1). Read normalization and pairwise differential gene expression analysis with

multiple testing corrections were conducted using the R Bioconductor DESeq2 package (Version 1.26). The heat map was produced by R (heatmap.2) using normalized RNA-seq read counts from a list of genes of interest. Genes reads counts per sample were first normalized by the 'Relative Log Expression' normalization (RLE) implemented in the DESeq2. Each normalized count was then normalized by the mean normalized value of the gene across all samples, in order to remove differences due to the level of expression of each gene. Samples are clustered in the left side of the figure, with a dendrogram, computed with euclidian distance and the complete agglomeration method. Raw data are available on Sequence Read Archive (SRA) database under the accession number PRJNA608249.

## Behavioral procedures

Behavioral tests were conducted as previously described in *Marty et al., 2016* and references therein. Most of them were performed blind during the light phase (from 8:30 a.m to 1 p.m.) using 3- to 5-month-old male mice (backcrossed on the C57BL/6J genetic background for n > 8 generations). **Open Field** - Locomotion and exploratory behavior were measured in a circular arena (height, 30 cm; diameter, 40 cm) located in a room containing no conspicuous features and illuminated by a white light (20 lx). Time spent in the center of the arena was recorded using the video tracker software (Ethovison XT, Noldus, Netherlands). Locomotion was scored during a 10 min session whereas center vs border activity were only analyzed during the first 5 min. **Elevated Plus Maze** - Mice were placed in an automated EPM (Imetronic, Pessac, France). This maze consisted of a plus (+)-shaped track with two closed and two open arms (30 × 10 × 20 cm) that extended from a central platform (10 × 10 cm). The apparatus was elevated 50 cm above the floor and was surrounded by a white curtain with no conspicuous cues. Each trial began with the mouse placed in the central zone. The number of entries into the closed and open arms and the time spent in each arm were monitored during the first 5 min. **Novelty-Suppressed Feeding -** Food-deprived mice for 24 hr were tested in a 50 × 50 × 20 cm box covered with bedding and illuminated by a 70 W lamp. A single pellet of food was placed on a white paper positioned in the center of the box. All mice were tested individually for 10 min and the latency time before the mouse ate the pellet was scored as a measure of motivation. Mice that did not eat during the testing period were not included in the analysis. Immediately afterwards, each single mouse returned to its home cage and the amount of food consumed was measured for 5 min (control for change in appetite as a possible confounding factor). **Tail suspension test -** Mice were suspended by their tails with tape, such that they cannot escape or hold on to nearby surfaces. The test was performed using the Tail Suspension Test System (BIOSEB) during a 6-min session. Immobility time was manually scored as an index of resignation. **Forced swim test -** Mice were introduced individually in a plastic cylinder (25 cm tall ×18 cm in diameter) filled with water (23–25°C) to a depth of 19 cm. A 6-min session was used and immobility time was only scored during the last 4 min. **Three-chamber test -** Social behavior was measured in a rectangular three-chambered enclosure (60 × 40 cm) with clear walls 22 cm high. Removable doors blocked access from the center chamber to the outer chambers. Two cages (diameter 8 cm, height 15 cm) were placed in the two outer chambers. On day 1 (5 min habituation session), mice were introduced into the apparatus with two empty cages. On day 2, a stranger mouse (S1) was introduced in one of the two cages and time spent interacting with S1 (sniffing) was measured during a 5 min session. Preference for S1 over the empty cage was used as a measure of sociability. During the second stage (same day), mice were submitted to two consecutive sessions with the same stranger with an inter-session resting period of 1 hr during which they were returned to the home-cage. On day 3, mice were submitted to the social novelty protocol (5-min session), with the previously encountered S1 mouse introduced in one cage and another stranger mouse (S2) was placed in the other cage. Preference for the previously encountered mouse over the familiar one was used as a measure of preference for social novelty.

## Determination of basal metabolism by indirect calorimetry

Indirect calorimetry was performed after 24 hr of acclimatization in individual cages. Oxygen consumption, carbon dioxide production, and food and water intake were measured (Phenomaster; TSE Systems, Bad Homburg v.d.H., Germany) in individual mice at 15 min intervals during a 24 hr period at constant temperature (22°C). The respiratory exchange ratio ([RER] = $V_{CO_2}/V_{O_2}$) was measured.

Energy expenditure (in kcal/day/kg$^{0.75}$ = 1.44 × $Vo_2$ × [3.815 + 1.232 × RER]), glucose oxidation (in g/min/kg$^{0.75}$ = [(4.545 × $Vco_2$) − (3.205 × $Vo_2$)]/1000) and lipid oxidation (in g/min/kg$^{0.75}$ = [1.672 × ($Vo_2$ − $Vco_2$)]/1000) were calculated. Ambulatory activities of the mice were monitored by infrared photocell beam interruption (Sedacom; Panlab-Bioseb).

## Glucose test tolerance and body composition

Glucose tolerance test (GTT) was performed under chow diet and at 12 weeks of HFD (i.p. and oral glucose (1.5 g/kg) administration, respectively) after an overnight fast and blood glucose levels from the tail vein were monitored over time using a glucometer (Accu-Chek). Total body fat and lean mass on chow diet were determined in mice placed in a clear plastic holder, without anaesthesia or sedation, and inserted into the EchoMRI-3-in-1 system (Echo Medical Systems, Houston, TX). Body composition was also evaluated by weighing liver, brown adipose tissue, perigonadal and subcutaneous fat.

## In situ hybridization on brain tissues

RNA FISH experiments were carried out on coronally sectioned 4% PFA fixed brain tissue (25 μm; cryostat) collected from adult *Pomc-eGFP* transgenic mice (C57BL/6J-Tg(Pomc-EGFP)1Low/J; Jackson Laboratory). Briefly, slides were hybridized overnight at 37°C in 15% formamide, 2X SSPE, 10% dextran sulfate, 150 μg/mL yeast tRNA and 10 ng of Cy3-labeled SNORD115 oligonucleotide probe (TXGAGCATGAATTTXATGTCATCACCTXTCTTCATGACAXT, with X = Amino-allyl-modified nucleotides). Samples were washed at room temperature in 15% formamide, 2X SSPE (20 min, twice), then 1xXSSPE (10 min) and mounted in Moviol DAPI (0.1 mg/mL). Images were captured with a CoolSnap HQ camera (Princeton Instruments) mounted on an inverted IX-81 microscope (Olympus) with a 1006 PL APO ON objective (NA 1.4) and the Metavue software.

## Statistical methods

R and the Graphpad Prism (version 8) statistical software were used for data analysis. For multivariate comparisons, statistical significance of the observed effects was calculated using ANOVA. When ANOVA applicability conditions were not met (variance heterogeneity or non-normality of residuals), measured values were log-transformed. In a few cases (*Figure 5E and F*, the two panels of *Figure 5L*, *Figure 6E* and *Figure 6G*), ANOVA applicability conditions were not fully met even after log-transformation, resulting in imprecise p-value calculation. For time-course analyses, explanatory variables were either genotype and time (*Figure 6A*) or genotype and a variable depending on time by a bi-exponential function, to account for the non-linearity of glucose concentration upon time (*Figures 5C* and *6C*). Data files and R scripts for these analyses have been deposited on https://github.com/HkeyHKey/Hebras_et_al_2020 (*Seitz, 2020*; copy archived at swh:1:rev:1b96b6f5e8-d479eb43f12f6687f04d4a60f4e305). For post-hoc tests and for univariate comparisons with two groups, significance was assessed using Student's t-test, or using the Mann-Whitney test when experimental values were not normally distributed. Data are presented as mean ± SEM and statistically significant differences are indicated as *p<0.05,**p<0.01, ***p<0.001 and ****p<0.0001.

## Acknowledgements

We thank the ABC Facility of ANEXPLO Toulouse for the mouse husbandry. We are grateful to the GeT core facility, Toulouse, France (get@genotoul.fr) and the Phenotypage service - US006/CREFRE INSERM/UPS. We are particularly indebted to Sophie Le Gonidec, Pauline Heuillard and Camille Eché for their technical assistance and scientific support. The authors also acknowledge «the Mouse Behavioral Core» (Center of Integrative Biology, Toulouse), notably Sébastien Lopez for his advices in setting up behavioral apparatus and procedures. We thank Xavier Fioramonti for the gift of PFA-fixed brains dissected from *Pomc*-eGFP transgenic mice. We also thank G Canal for assistance in the high-throughput analysis of A-to-I RNA editing. This work was supported by the 'Fondation pour la Recherche Médicale' (FRM; DEQ20160334936), the Agence Nationale de la Recherche (ANR-18-CE12-0008-01) and the Fundation for Prader-Willi Research (FPWR). J Hebras is grateful to the 'La ligue Nationale contre le Cancer' for the 4th year PhD funding.

## Additional information

### Funding

| Funder | Grant reference number | Author |
|---|---|---|
| Foundation for Prader-Willi Research | | Jérôme Cavaille |
| Fondation pour la Recherche Médicale | DEQ20160334936 | Jérôme Cavaille |
| Agence Nationale de la Recherche | ANR-18-CE12-0008-01 | Jérôme Cavaille |

The funders had no role in study design, data collection and interpretation, or the decision to submit the work for publication.

### Author contributions

Jade Hebras, Conceptualization, Formal analysis, Validation, Investigation, Writing - review and editing; Virginie Marty, Formal analysis, Validation, Investigation, Writing - review and editing; Jean Personnaz, Pascale Mercier, Nicolai Krogh, Formal analysis, Investigation, Writing - review and editing; Henrik Nielsen, Supervision, Writing - review and editing; Marion Aguirrebengoa, Resources, Data curation, Software, Visualization, Writing - review and editing; Hervé Seitz, Conceptualization, Resources, Data curation, Software, Validation, Visualization, Writing - review and editing; Jean-Phillipe Pradere, Formal analysis, Supervision, Investigation, Writing - review and editing; Bruno P Guiard, Formal analysis, Supervision, Writing - review and editing; Jérôme Cavaille, Conceptualization, Supervision, Funding acquisition, Writing - original draft, Project administration, Writing - review and editing

### Author ORCIDs

Jean Personnaz [iD] http://orcid.org/0000-0001-6447-780X
Nicolai Krogh [iD] https://orcid.org/0000-0001-8870-7091
Henrik Nielsen [iD] https://orcid.org/0000-0001-7143-2810
Hervé Seitz [iD] http://orcid.org/0000-0001-8172-5393
Jérôme Cavaille [iD] https://orcid.org/0000-0003-2833-6836

### Ethics

Animal experimentation: Animal procedures were approved by the University of Toulouse and CNRS Institutional Animal Care Committee (DAP2016061716367988 and DAP2018011214542827). The animal housing facility met CNRS standards.

### Decision letter and Author response

Decision letter https://doi.org/10.7554/eLife.60862.sa1
Author response https://doi.org/10.7554/eLife.60862.sa2

## Additional files

### Supplementary files

• Supplementary file 1. RiboMeth-seq scores and statistics. Raw data are deposited at NCBI Gene Expression Omnibus (GEO) and accessible through GSE145159. Note that RiboMeth-seq data from WT littermates were also included in *Hebras et al., 2020* RNA biol.

• Supplementary file 2. *Snord115* is expressed in pro-opiomelanocortin (*Pomc*) neurons in the arcuate nucleus of the hypothalamus. Coronally sectioned brain tissues of adult P*omc*-eGFP transgenic mice (*Pomc* neurons appear green in the merge) were hybridized with Cy3-labeled DNA oligonucleotides (red signals in the merge). Bottom: *Snord115* signals are detected in nuclear regions poorly stained by DAPI (white arrow), very likely representing nucleoli.

- Supplementary file 3. Changes in transcript steady-state levels in the adult hypothalamus of *Snord115*-deficient mice relative to WT littermates (mRNA-seq). Raw *data* are available on Sequence Read Archive (*SRA*) database under the accession number PRJNA608249.

- Supplementary file 4. Sequences of DNA oligonucleotides used in the study.

- Transparent reporting form

## Data availability

RiboMeth-seq data are available on GEO under the accession number GSE145159 Raw data (mRNA-seq) are available on Sequence Read Archive (SRA) database under the accession number PRJNA608249. Raw data (A-to-I RNA editing) are available on Sequence Read Archive (SRA) database under the accession numbers PRJNA603261 and PRJNA603264. Scripts used for that analysis, detailed instructions and intermediary data have been deposited at https://github.com/HKeyHKey/Hebras_et_al_2020 (copy archived at https://archive.softwareheritage.org/swh:1:rev:1b96b6f5e8-d479eb43f12f6687f04d4a60f4e305/).

The following datasets were generated:

| Author(s) | Year | Dataset title | Dataset URL | Database and Identifier |
|---|---|---|---|---|
| Hebras J, Marty V, Personnaz J, Mercier P, Krogh N, Nielsen H, Aguirre-bengoa M, Seitz H, Pradère J-P, Guiard BP, Cavaillé J | 2020 | Re-assessment of the involvement of Snord115 in the serotonin 2C receptor pathway in a genetically relevant mouse model | http://www.ncbi.nlm.nih.gov/geo/query/acc.cgi?acc=GSE145159 | NCBI Gene Expression Omnibus, GSE145159 |
| Hebras J, Marty V, Personnaz J, Mercier P, Krogh N, Nielsen H, Aguirre-bengoa M, Seitz H, Pradère J-P, Guiard BP, Cavaillé J | 2020 | Measurement of A-to-I editing in various adult mouse brain areas in SNORD115-expressing and SNORD115-deficient specimens | https://www.ncbi.nlm.nih.gov/bioproject/PRJNA603261 | NCBI BioProject, PRJNA603261 |
| Hebras J, Marty V, Personnaz J, Mercier P, Krogh N, Nielsen H, Aguirre-bengoa M, Seitz H, Pradère J-P, Guiard BP, Cavaillé J | 2020 | Measurement of A-to-I editing in various embryonic, neonate and adult mouse brain areas in SNORD115-expressing and SNORD115-deficient specimens | https://www.ncbi.nlm.nih.gov/bioproject/PRJNA603264 | NCBI BioProject, PRJNA603264 |
| Hebras J, Marty V, Personnaz J, Mercier P, Krogh N, Nielsen H, Aguirre-bengoa M, Seitz H, Pradère J-P, Guiard BP, Cavaillé J | 2020 | Re-assessment of the involvement of SNORD115 in the serotonin 2C receptor pathway in a genetically relevant model in vivo | https://www.ncbi.nlm.nih.gov/bioproject/PRJNA608249 | NCBI BioProject, PRJNA608249 |

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
