## [Decision Letter]

Congratulations, we are pleased to inform you that your article, "Re-assessment of the involvement of *SNORD115* in the serotonin 2C receptor pathway in a genetically relevant mouse model", has been accepted for publication in *eLife*.

Reviewer #1:

The manuscript by Hebras et al. from the Cavaille laboratory describes an interesting investigation of the disruption of the *SNORD115* gene region in a mouse model using CRISPR/Cas9-dediated knockout of *SNORD115* and the role that such a mutation has on behavioral phenotypes in the mouse model. Deletions in the *SNORD115* regions have been reported in patients diagnosed with Prader-Willi syndrome (PWS). The human studies for this disease as well as other previous related studies in mice reported correlations of various mutations involving the *SNORD115* gene with impaired emotional response and/or compulsive overeating phenotypes. Importantly, Dr. Cavaille's laboratory had been one of the laboratories reporting on this correlation in mice. However, in this study Dr. Cavaille's laboratory contradicts the results of these studies and then proceeds to explain the underlying causes for this alternative interpretation.

These underlying causes include the likely effects of using cultured non-neuronal cell lines as well as brain tissues (whole brain or hypothalamus) in studying the alternative splicing of 5-HT2C primary mRNA transcripts by SNORD115. Another cause that was posited are the genotypes of previously used mouse lines (PWS-IC and the PatDp?+). These lines contain pre-exiting large genomic lesions compared to the CRISPR/Cas9 engineered lines used in their study. These differences are certainly the possible reasons for the disparity in the results reported previously and in this study.

Thus, these observations should be available to the community who have studied this genotype-phenotype relationship. However, the most compelling point made by the authors in explaining the contradiction is made at the end of the Discussion section. The authors' observation that the role of *SNORD115* is complex and that it may be only one of multiple regulators is reasonable and would prove to be less contentious to those who may favor the former explanations of the cause of the behavioral phenotypes. It would seem that a statement to this effect should be made more prominently early in the report rather than requiring the reader to wait until the last few lines of the paper.

Reviewer #2:

While the genetic aetiology of Prader Willi Syndrome (PWS), the loss of expression of a repertoire of imprinted genes on Ch15q, has been known for many years, the mechanistic underpinnings of the disease have remained elusive. Two reasons for this, the large number of genes in the region, and the fact that mouse models of PWS do not recapitulate the full phenotypic spectrum of the disease, including hyperphagia and obesity. In recent years, attention has been focussed on two clusters of SNORD RNAs in the region, *SNORD115* and *SNORD116*. There is now a critical mass of evidence showing that humans carrying micro-deletions encompassing *SNORD116* display much of the phenotype of PWS. However, there has been high-profile and controversial work that argues a role of *SNORD115*, via its role in the splicing of the serotonin receptor 5HT-2C.

Here Jade and colleagues provide an important and rigorously performed piece of work that argues for a reassessment of the hypothesis that *SNORD115* plays a key role in PWS. The work covers all of the important PWS characteristics and, in brief, shows that genetic deletion of *SNORD115* in mice results in phenotypes indistinguishable from wild-type littermates, and critically that splicing of 5HT-2C is not significantly altered.

I have no substantive concerns to raise.

---

## [Author Response]

Reviewer #1:[…]Thus, these observations should be available to the community who have studied this genotype-phenotype relationship. However, the most compelling point made by the authors in explaining the contradiction is made at the end of the Discussion section. The authors' observation that the role of SNORD115 is complex and that it may be only one of multiple regulators is reasonable and would prove to be less contentious to those who may favor the former explanations of the cause of the behavioral phenotypes. It would seem that a statement to this effect should be made more prominently early in the report rather than requiring the reader to wait until the last few lines of the paper.

As recommended by this reviewer, we slightly reshaped a sentence in the Introduction as follows:

“Yet, direct proofs supporting this appealing *SNORD115*/HTR2C axis are still lacking at the organism level (8). As a corollary, whether lack of SNORD115 per se is sufficient to trigger altered HTR2C-signalling pathways and, in doing so, leads to behavioural abnormalities remains to be formally demonstrated”.”